# FLIMJ: An open-source ImageJ toolkit for fluorescence lifetime image data analysis

Dasong Gao[1☯], Paul R. Barber[2☯], Jenu V. Chacko[1], Md. Abdul Kader Sagar[1,3], Curtis T. Rueden[1], Aivar R. Grislis[1], Mark C. Hiner[1], Kevin W. Eliceiri[1,3,4,5]*

**1** Laboratory for Optical and Computational Instrumentation, Center for Quantitative Cell Imaging, University of Wisconsin, Madison, WI, United States of America, **2** UCL Cancer Institute, Paul O'Gorman Building, University College London, London, United Kingdom, **3** Department of Biomedical Engineering, University of Wisconsin, Madison, WI, United States of America, **4** Department of Medical Physics, University of Wisconsin, Madison, WI, United States of America, **5** Morgridge Institute for Research, University of Wisconsin, Madison, WI, United States of America

☯ These authors contributed equally to this work.
* eliceiri@wisc.edu

**Data Availability Statement:** All relevant data used in this manuscript are provided in the SCIFIO sample datasets repository accessible at https://scif.io/images. The specific FLIM dataset can be

## Abstract

In the field of fluorescence microscopy, there is continued demand for dynamic technologies that can exploit the complete information from every pixel of an image. One imaging technique with proven ability for yielding additional information from fluorescence imaging is Fluorescence Lifetime Imaging Microscopy (FLIM). FLIM allows for the measurement of how long a fluorophore stays in an excited energy state, and this measurement is affected by changes in its chemical microenvironment, such as proximity to other fluorophores, pH, and hydrophobic regions. This ability to provide information about the microenvironment has made FLIM a powerful tool for cellular imaging studies ranging from metabolic measurement to measuring distances between proteins. The increased use of FLIM has necessitated the development of computational tools for integrating FLIM analysis with image and data processing. To address this need, we have created FLIMJ, an ImageJ plugin and toolkit that allows for easy use and development of extensible image analysis workflows with FLIM data. Built on the FLIMLib decay curve fitting library and the ImageJ Ops framework, FLIMJ offers FLIM fitting routines with seamless integration with many other ImageJ components, and the ability to be extended to create complex FLIM analysis workflows. Building on ImageJ Ops also enables FLIMJ's routines to be used with Jupyter notebooks and integrate naturally with science-friendly programming in, e.g., Python and Groovy. We show the extensibility of FLIMJ in two analysis scenarios: lifetime-based image segmentation and image colocalization. We also validate the fitting routines by comparing them against industry FLIM analysis standards.

## Introduction

In the last thirty years, numerous advanced biological imaging techniques have allowed for the interrogation of biological phenomena at cellular and subcellular resolution. One of these

accessed directly at https://samples.scif.io/Gray-FLIM-datasets.zip. The GitHub repository for the library and example notebooks is available at https://flimlib.github.io.

**Funding:** We acknowledge support from NIH grants R01CA185251 (KWE), RC2 GM092519 (KWE), P41GM13501 (KWE), the Semiconductor Research Corporation (KWE), U.S. Department of Energy grant DE-SC0019013 (KWE), and the Morgridge Institute for Research (KWE). This work was also supported by the Cancer Research UK Programme grant C5255/A15935 (PRB) the CRUK UCL Centre grant C416/A25145 (PRB), CRUK City of London Centre grant C7893/A26233 (PRB), CRUK KCL-UCL Comprehensive Cancer Imaging Centre (CRUK & EPSRC) and MRC and DoH grants C1519/A16463 and C1519/A10331 (PRB). We also thank the UW-Madison Hilldale Undergraduate/ Faculty Research Fellowship (DG and KWE) and UW-Madison Trewartha Senior Thesis Research Grant program (DG) for their support. The funders had no role in study design, data collection and analysis, decision to publish, or preparation of the manuscript.

**Competing interests:** The authors have declared that no competing interests exist.

powerful techniques is modern fluorescence microscopy, empowered by key inventions such as the laser scanning microscope and the use of fluorescent proteins. Among the range of modalities, Fluorescence Lifetime Imaging Microscopy (FLIM) has been of particular interest in molecular imaging diagnostics due to its ability to probe the cellular microenvironment, sensitivity to changes in molecular conformations, and utility in interpreting phenomena such as Förster Resonance Energy Transfer (FRET) and physiological states including pH and hydrophobicity. FLIM is now widely used in a variety of cell imaging applications, from measuring the metabolic state of differentiating stem cells [1] and intrinsic signatures of cancer cells [2] to measuring changes in lipid rafts [3] and FRET of signaling events in cell division [4]. FLIM is available in two primary modes of operation, time-domain [5] and frequency domain [6], but also is compatible with a number of different microscopy configurations, including wide-field [7], confocal [8], spinning disc [9], and multiphoton [10] microscopy. Several research groups have demonstrated super-resolution FLIM [11] and medical applications of FLIM, including ophthalmology and endoscopy [12]. These emerging applications of FLIM continuously push further the innovation of FLIM technology including faster electronics and more sensitive detection. Despite these advances in biological applications and instrumentation, there has been a surprising lack in corresponding development in image informatics tools to directly support the FLIM imaging and analysis. Specifically, as a quantitative technique that generates image datasets, FLIM has the inherent need for powerful downstream image analysis software to interpret the results.

While more work is needed, many recent advances in FLIM have largely been enabled by improved computation and software. Advanced software tools have not only allowed for FLIM electronics to be robustly controlled and capture short lifetimes, but to do so rapidly so that FLIM images in 3D (space) and 4D (space and time, or space and spectral) can be collected [13]. Curve fitting algorithms have been developed that allow for robust fitting of two or more components. Several companies have made commercial packages for FLIM analysis, but these are closed source tools that are not transparent in their analyses and typically only support their own file formats. This makes the sharing of approaches and FLIM data difficult while also limiting the usage of the features supported by the software. In recognition of the need for more transparent and customizable methods for FLIM analysis, there are new developments for turnkey open analysis FLIM software tools such as FLIMfit from Paul French's group [14]. However, most of these software packages are not designed with the rationale that FLIM results should be treated as images that can be segmented, statistically analyzed, or learned by upcoming newer machine learning algorithms. The separation of FLIM from other image analysis workflows has placed difficulty for biologists who would otherwise benefit from an easier image-based integration of FLIM.

To summarize, the scientific community would benefit from a more complete informatics approach addressing three specific and currently unmet needs:

1. an open and extensible FLIM algorithm library that supports the most popular FLIM file formats and can be utilized and modified easily by a developer;

2. a turnkey FLIM analysis tool that uses the library and yet still can be used easily and directly by the bench biologist; and

3. the integration of FLIM analysis with versatile microscopy image analysis.

To address the first two needs, Barber et al. developed the Time-Resolved Imaging version 2 (TRI2), a freely available closed source FLIM analysis application equipped with a LabWindows graphical user interface (GUI) and basic image analysis capabilities [15]. TRI2 was released to selected researchers in 2004, and from it, the core fitting algorithms were extracted to form the open-source FLIMLib library.

The remaining need can be fulfilled by exploiting established open-source image analysis platforms. A great platform to integrate FLIM analysis is the open-source ImageJ [16] and its distribution for the life sciences, Fiji [17]. ImageJ has long enjoyed use and adaptation by experimental biologists. However, FLIM workflows have largely been segregated from ImageJ due to a lack of the necessary FLIM analysis functionality. Fortunately, the recent development of the ImageJ Ops framework [18] has laid a solid foundation for such a FLIM analysis toolbox. As a backbone of the next-generation ImageJ2 that powers Fiji, the ImageJ Ops framework supports the growth of ImageJ's image processing and analysis power. In addition to offering hundreds of easy-to-use, general and efficient image processing operations such as segmentation algorithms, statistics, and colocalization methods across hundreds of file types, this framework provides programming interfaces for developers to extend the library. For example, the Ops-based plugins naturally cater to the needs of both bench biologists and advanced developers by allowing the former to easily use the Ops and the latter to build user-tailored applications for their own needs. Together with the seamless connections between Ops, the ImageJ Ops framework creates a suitable environment for developing modular, extensible image processing workflow components.

There were previous efforts by our group to integrate FLIM analysis into ImageJ, known as the SLIM Curve (Spectral Lifetime Imaging Microscopy) plugin for ImageJ, which provided a graphical interface to FLIMLib. The SLIM Curve plugin was able to integrate FLIMLib's full fitting functionalities with ImageJ while being accessible to bench biologists. However, the SLIM Curve plugin was built on legacy ImageJ 1.x data structures, which limited its extensibility and modularity compared to the ImageJ2 infrastructure.

We now have built on previous efforts in lifetime analysis [15, 19] and spectral lifetime analysis [20] to build an ImageJ-centric toolkit, FLIMJ, that directly addresses these three needs. A schematic representation of the FLIMJ framework is presented in Fig 1. FLIMJ is an international collaboration between software developers and microscopists at the UCL Cancer Institute, London, UK (and formerly at the Gray Institute for Radiation Oncology and Biology at the University of Oxford, UK) and the UW-Madison Laboratory for Optical and Computational Instrumentation (LOCI) in the USA that leverages several existing software projects. When designing this FLIM analysis system, we recognized that current methods, such as those in MATLAB, may be difficult for the biological community to use. Further, many of these methods are also not attractive for developers because much of the published code, such as Numerical Recipes [21], may have restrictive licenses and a steep learning curve. We sought to develop a toolkit that would complement current commercial efforts and in fact, directly support the image acquisition and image processing formats of these systems. We decided to focus our initial efforts on time-resolved FLIM data collected from the PicoQuant and Becker & Hickl hardware systems, but the toolkit is designed to be flexible enough to support frequency or spectral domain analyses. Currently, spectral support is minimal, but this will be augmented as the toolkit evolves. This flexibility is supported by the use of the ImgLib2 (http://imglib2.net/) data container as widely adopted by almost all of the ImageJ plugins, which allows for the support of data of, in principle, unlimited dimensions and will handle FLIM data that includes additional channels of spectra and polarization.

As described below, the FLIMJ toolkit can either be invoked as an ImageJ Op or used from the graphical user interface (GUI) equipped plugin. One powerful advantage ImageJ can offer in FLIM analysis is segmentation. Looking at the lifetime distribution of a specific manually defined ROI can often be cumbersome or not even possible with commercially available FLIM analysis software packages. ImageJ not only has conventional manual segmentation but in addition, supports machine learning-based classification techniques [22], which can be exploited to use a training set to segment images in batch processing automatically. A number

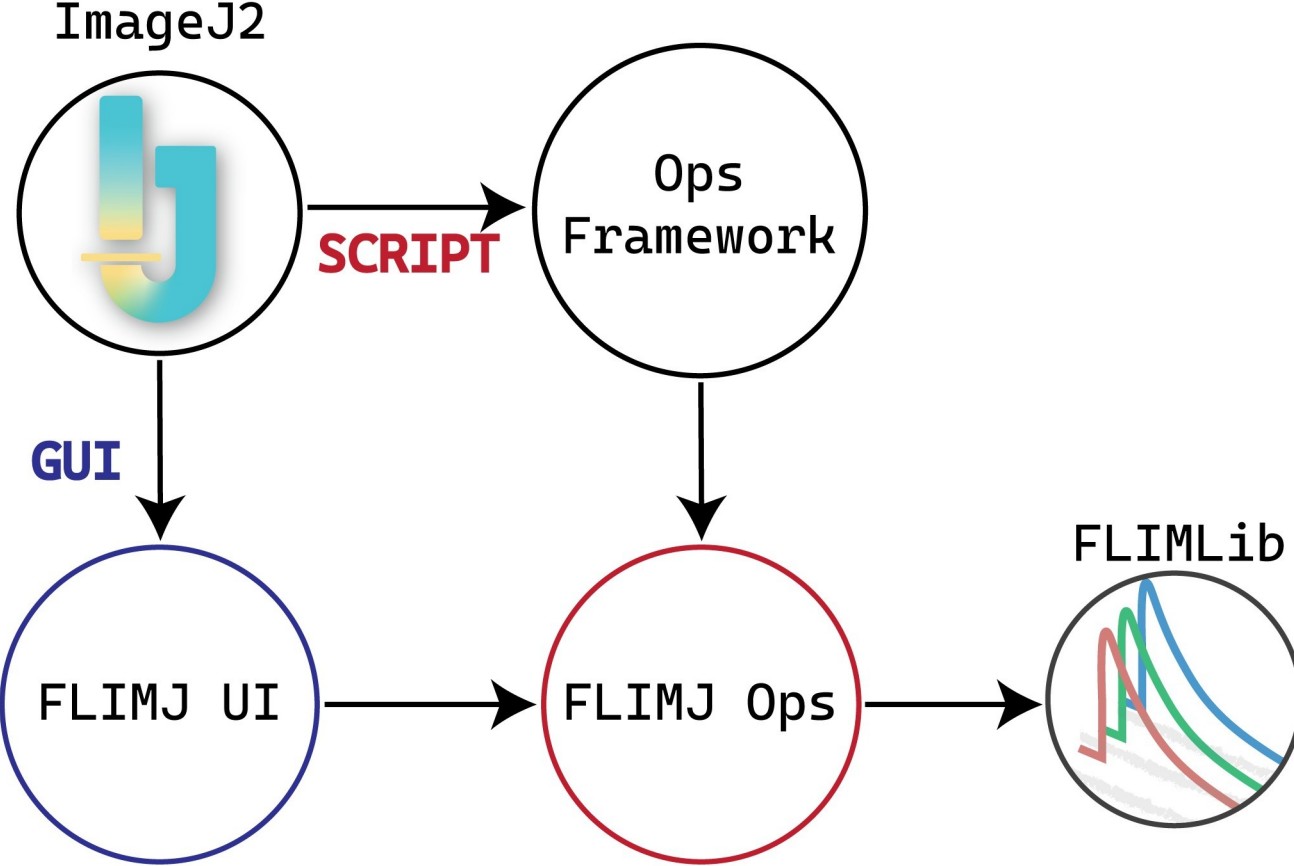

**Fig 1. Relationships between components of FLIMJ and ImageJ2.** FLIMJ Ops depends on FLIMLib and communicates with other supporting ops (mathematical, statistical, and input/output tools) through the Ops framework. This schematic shows two different ways to access FLIMLib. The scripting path goes through the ops-framework to the FLIMJ-Ops library, and the GUI path goes through the FLIMJ-UI to FLIMJ-Ops.

of other ImageJ features could be utilized with FLIM data, including scripting, 3D visualization, and feature tracking, as well as a myriad of imaging file formats that can be supported through the use of the SCIFIO (http://scif.io/) infrastructures [23] to support a range of open and proprietary FLIM file formats. To our knowledge, this is the first FLIM analysis system that offers this range of flexibility and functionality.

The approach to use a core open-source software library brings the usual advantages of open-source that enable community contribution to and verification of the underlying code. It also enables the release of a 'polished' graphical-user-interface-driven plugin and allows use in other environments. In the following section, a description of the toolkit is given, followed by how it can be extended for some more-advanced uses.

## Methods

FLIMJ provides access to a variety of FLIM analysis techniques including standard nonlinear least-squares fitting in the form of the Levenberg-Marquardt (LM) algorithm and more advanced algorithms such as maximum likelihood, global, and Bayesian analysis that has been optimized for FLIM [15], as well as simpler methods such as the rapid lifetime determination (RLD) by integration [24], and frequency domain analysis via the method of phasors [25, 26]. In particular, the toolkit has the ability to account for an instrument response function (IRF,

prompt, or excitation function) that distorts the pure exponential decay. Through iterative reconvolution, the LM algorithm extracts the true lifetime estimate from IRF distorted signals. The addition of new methods is also allowed and can inherit the standard library code interface as exemplified by the recently incorporated Bayesian algorithms [27]. Similar to any other open or proprietary FLIM analysis software, FLIMJ uses FLIM datasets of any accessible format (supported by Bio-Formats [28]) and outputs the results as images for further examination. Using ImageJ's high-dimensional image data structure for both the input and output, FLIMJ ensures maximal compatibility with other parts of ImageJ, which can provide powerful preprocessing and post-analysis for the FLIM workflow (see Results).

The toolkit employs a modular design and comprises three major components: FLIMLib, FLIMJ Ops, and FLIMJ-UI. FLIMLib is the underlying library that contains an efficient C implementation of the algorithms. Based on the ImageJ Ops framework, FLIMJ Ops implements adapter ops for each of the algorithms in Java to dispatch the input data, invoke the corresponding C routines in FLIMLib, and collect the results. While FLIMJ Ops, together with the underlying FLIMLib, deliver core functionality programmatically, FLIMJ-UI greatly improves the accessibility for bench biologists by providing an intuitive GUI based on the JavaFX framework to allow for the visualization and fine-tuning of the fit. In the rest of the section, we present a detailed description of each of the components.

## FLIMLib

FLIMLib is a cross-platform compatible library written in ANSI C with a Java API extension. With help from the current Maven-CMake building mechanism, the library can be compiled to run as a native executable on Linux, Windows, or macOS. As the weight-lifting component of FLIMJ, FLIMLib is equipped with a Java Native Interface (JNI) wrapper created by the SWIG framework (http://swig.org/), which offers efficient type conversion and data transfer between C and Java applications. However, more connectors can be added to make the library accessible to many high-level programmers using Python, MATLAB, C++, and C#, to name a few. Besides integrations with other languages, the library can also be compiled into a standalone command-line program with the intention that user interaction with the library could be in several forms according to the user's choice. That is, the interaction could be through a graphical user interface such as with TRI2 or ImageJ, or could be through the command line in a scriptable form, or could be via a third-party framework such as Python (http://www.python.org), MATLAB (www.mathworks.com) or R (http://www.r-project.org/). Full details on FLIMLib, including downloads, can be found on the project web site at https://flimlib.github.io.

The following fitting and analysis methods are currently present in the open-source library for lifetime data:

- **Rapid Lifetime Determination (RLD):** A fast computation method based on three integrals to determine a single average lifetime [24], including a variant that accounts for the IRF [24, 29].

- **Levenberg-Marquardt (LM) Non-Linear Least Squares Fitting:** A classical LM algorithm [30], the performance of which is modified by the noise model. Multi-exponential and stretched exponential analyses are built-in, others can be added, as are parameter fixing and restraining. There are variants with and without an IRF. Parameter error estimates are returned based on the fitting alpha matrix [21]. Possible noise models include Maximum Likelihood Estimation: Poisson noise model [31] and Gaussian: Variance for Gaussian distribution (Least-squares fitting). These statistical noise models used here were described in detail in Rowley's work [27].

◦ *Maximum Likelihood Estimation*: This variant of the LM algorithm is accessed by using a specific noise model. Several noise models can be chosen to influence the LM optimization via the chi-squared ($\chi^2$) parameter. The models are a) '*Constant*'—every data point is assumed to have the same supplied variance. b) '*Given*'—every data point can have an individual variance, given via a data array. c) '*Gaussian*'—Variance for Gaussian distribution is used at all data points. d) '*Poisson*'—Variance for Gaussian distribution is used with a lower limit of 15, this being the point where the Gaussian approximation begins to break down with Poissonian data. e) '*MLE*'—Maximum likelihood estimation through the use of the Poisson equation [31].

- *Global Analysis*: In some biological experiments, it is extremely advantageous to analyze data from a whole image or a series of images simultaneously to determine certain parameters with a high accuracy yet allow other parameters to remain variable at the pixel level to capture spatial variations. In this so-called 'global analysis,' some parameters can be globally determined while others remain local [32, 33]. This method is particularly useful when a biosensor based on FRET is in use [34], which usually exists in either an activated or deactivated state that is represented by different measured lifetimes by FLIM. Thus, these characteristic lifetimes can be determined globally, whilst the determination of the fraction of activated molecules can be determined locally [15]. The library has built-in optimized functions for this type of analysis [19], that offer fast convergence and built-in optimization of initial parameter estimates. Methods for global analysis involving other generic functions (e.g., an exponential rise or non-exponential functions) are also included in the library.

- *Phasor*: Transformation to phasor space for the calculation of a single average lifetime and graphical multi-exponential analysis [26].

- *Bayesian Inference*: Lifetime estimation based on Bayesian inference offers higher precision and stability when faced with data with low photon counts (low signal-to-noise ratio). This algorithm acts by combining evidence from the photon arrival times to produce robust estimates of lifetimes and the potential errors in those estimates. In in-vitro experiments, it was found that the precision was increased by a factor of two compared to LM fitting, or acquisition time could be reduced by a factor of two for an equivalent precision [27]. The algorithms in the library can be used to estimate the IRF and exponential decay simultaneously or can be used to perform model selection between mono- and bi-exponential fitting models.

## FLIMJ Ops

FLIMJ Ops is a plugin built upon the ImageJ Ops framework [18, 35] that connects FLIMLib and the ImageJ ecosystem. Based on the ImageJ Ops template, the plugin adapts the single-transient RLD, LMA, Global, Bayesian, and phasor analysis functionalities from FLIMLib into dataset-level fitting ops accessible from ImageJ. With help from the ImageJ Ops framework, FLIMJ Ops provides a concise yet flexible programmatic interface that can be easily included in a scripting workflow (see Results).

## Ops API

Conforming to the organization convention of ImageJ Ops, the fitting ops are contained in the **flim** namespace. Specifically, by calling **flim.fit***⁎* with *⁎* being **RLD**, **LMA**, **Global**, **Bayes** or **Phasor**, the user performs the corresponding analysis implemented by FLIMLib over the dataset on each individual pixel. While fully preserving FLIMLib's granularity of control

over fitting parameters such as noise model, initial estimation, and IRF, the Ops API provides support for dataset-level preprocessing operations, including pixel binning and ROI cropping. To optimize for the ease of use and prevent the need for a verbose argument list, some of the op arguments are marked as optional **(required = false)** at definition so that the programmer can ignore them in op calls. Upon invocation by name, the ImageJ Ops framework automatically inspects the number and types of arguments passed in, matches the appropriate fitting op, and completes the argument list by assigning sensible default values to optional parameters.

FLIM dataset analysis tasks greatly benefit from the parallelized fitting of individual pixels if they can be analyzed independently. This paradigm is employed by TRI2 and SPCImage by leveraging either the central processing unit (CPU) multithreading or graphics processing unit (GPU) acceleration. However, in terms of portability, CPU multithreading is considered more favorable. So far, all fitting ops in FLIMJ Ops, except for the global analysis op, by default run on multiple CPU threads in parallel. The parallelization relies on the ImageJ **ChunkerOp** utility for dispatching the workload.

## FLIMJ-UI

Oftentimes, it may be desirable for analytical software to integrate tools for visualizing results and to allow fine-tuning of the configurations. This is especially true for FLIM applications since the fitting results are usually sensitive to the settings, and manual setting of parameters such as initial values and decay interval range is required to obtain the optimal fit. The FLIMJ-UI is an ImageJ plugin created for such needs. Based on the SciJava Command framework, the plugin invokes FLIMJ Ops to carry out the computation and displays the fitted parameters alongside the decay curve with a JavaFX GUI. Like any of the ImageJ Ops, the plugin can be started through scripting, or the user may launch it from Fiji during an image analysis workflow.

## Notebooks for FLIM analysis

Notebooks are scientific programming tools that bind the processing pipeline (codes) with the input and output assisted with detailed comments using markdown-style notes. Experiments and Image datasets can now be associated with processing routines pointing out dependencies and programming environments used for extracting results. Currently, notebooks are available to most scientific languages, including Mathematica, MATLAB, Python, R, Julia, Groovy, and others under tag names of Jupyter, BeakerX, and Zeglin notebooks. We supplement the FLIMJ Ops with a Groovy and Python notebook that can help a beginner to use fitting FLIM data in an interactive and data-analysis friendly way. Two example demo notebooks are provided with the FLIMJ Ops repository: 1) a groovy notebook running on the BeakerX kernel that accesses ImageJ ops directly, 2) a python Jupyter notebook that accesses ImageJ ops through the PyImageJ interface to invoke FLIMJ ops.

## Results

In this section, we demonstrate FLIMJ workflow and validate the results using separate software and simulated data. Two use-case scenarios are presented for image-segmentation and image-colocalization. a) Segmentation: FRET efficiency calculation of segmented tumorous tissue, b) colocalization of NAD(P)H, and antibody distribution for microglia. FLIMJ is shown to collaborate seamlessly and performantly with the central components of each workflow, as well as to have the potential to be integrated with more complex ones.

## Use case I: Segmentation

This use case is based on the FLIMJ plugin for ImageJ, and relates to linking fluorescence life-time processing to other advanced image processing plugins within ImageJ to accurately measure protein dimer information in cancer tissue [36]. Many cancers are driven by the dimerization of the members of the HER/ErbB protein family (EGFR, HER2, HER3 etc.) at the cell surface. This discovery has led to the invention of targeted therapies that specifically disrupt the ability of these proteins to dimerize, and with some success; e.g., a monoclonal antibody for EGFR, cetuximab, is now used alongside chemotherapy for certain colorectal and head & neck tumors, and trastuzumab is used to target HER2 in some breast cancers [37]. We have recently published the use of FLIM/FRET to detect HER family dimerization in archived patient tissue from breast and colorectal cancer studies and clinical trials [36, 37] and correlate this with the response to targeted treatment.

One challenging aspect of these measurements in tissue is the need to segment the tumor tissue from surrounding stroma and other normal tissue components. This can be achieved through multi-exponential lifetime filtering [38], but it is often better to use independent information provided by the tissue morphology provided by imaging to segment the tumor. In ImageJ, we have created a pipeline that uses trainable Weka segmentation [22] on the intensity image in parallel to FLIMJ to provide tumor segmented lifetime statistics. These results from two serial sections, stained with the FRET donor and acceptor, and donor alone as a control, provide a FRET measure of protein dimerization. Additionally, training can also be done on lifetime images post-processing to help identify dimmer species. Fig 2 shows representative sections of breast cancer tissue from the METABRIC study [37] and calculated FRET scores of HER2-HER3 dimerization.

Samples were imaged on a customized "open" microscope automated FLIM system [38]. Time-domain fluorescence lifetime images were acquired via time-correlated single-photon counting (TCSPC) at a resolution of 256 by 256 pixels, with 256-time bins and 100 frames accumulated over 300 seconds, via excitation and emission filters suitable for the detection of Alexa546 fluorescence (Excitation filter: Semrock FF01-540/15-25; Beam Splitter: Edmund 48NT-392 30R/70T; Emission filter: Semrock FF01-593/40-25). For technical convenience, those FLIM images were acquired through the emission channel of a UV filter cube (Long pass emission filter > 420 nm).

The pipeline for Use Case 1 can all be performed using the graphical user interface as follows. Total FLIM intensity images were created using the Z-Project function and these were used to train the Weka Trainable Segmentation plugin. The Weka Apply Classifier function then created segmented images of tumour areas which could be thresholded, and made into a mask and an ImageJ selection (Threshold, Create Mask and Create Selection functions). In parallel the FLIM image was analysed in the FLIMJ plugin using a mono-exponential LMA model. The export function from FLIMJ created a Tau image onto which the selection from the Weka output was applied (Restore Selection function). The Measure function was used to create the mean lifetime for the tumour area and images for display were created by applying the mask to the Tau image (Image Calculator min value between Tau and Mask). Once the Weka segmentation is trained, all these functions can also be scripted for automated processing.

## Use case II: Colocalization

This use case is based on the Fiji ROI colocalization plugin and links to fluorescence lifetime processing of autofluorescence images of microglia. Previously we and others have demonstrated the potential applicability of NAD(P)H FLIM in differentiating microglia functional state [39–41] and computational approaches to distinguish microglia cells [42]. Based on our

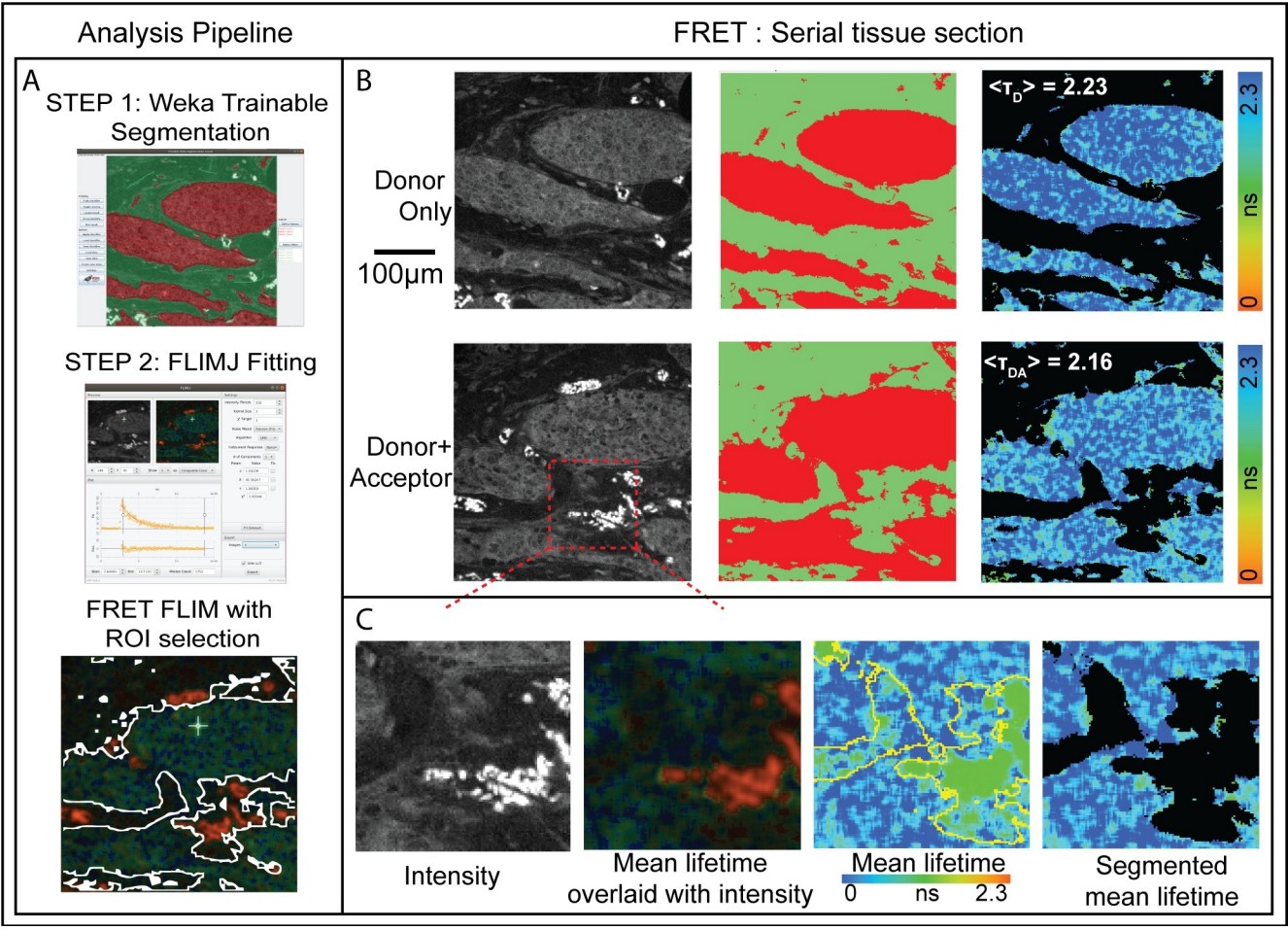

**Fig 2.** A) Breast cancer tissues from 107 patients METABRIC study were stained with antibodies: anti-HER3-IgG-Alexa546 (donor) and anti-HER2-IgG-Cy5 (acceptor) [37]. Serial sections were stained with donor+acceptor (DA, FRET pair) and with donor alone (D, control). Average lifetime values ($T_D$ and $T_{DA}$) can be determined for the tumor from the two serial sections using FLIMJ after segmentation. The FRET efficiency can be calculated according to FRETeff = 1 −TDA/TD. B) Weka Trainable Segmentation plugin was used to segment the tissue areas. The FLIMJ user interface showing a typical transient and fit from the tissue. We used the LM fitting with a mono-exponential model. C) Zoom into a smaller region. Composite image from FLIMJ showing lifetime information. Pure lifetime map with Weka segmentation shown in yellow. Segmentation result of the lifetime within the tumor with artifactual tissue removed. From $T_D$ = 2.23 ns and $T_{DA}$ = 2.16 ns, we estimate a FRET efficiency for this example tumor area as 3.1% as a measure of HER2-HER3 dimerization on the tumor in this patient.

findings in the previous works, we expect that the hybrid method allowing lifetime estimation from raw decay data and subsequent colocalization analysis can be helpful in determining the effectiveness of FLIM based approaches in the identification of a specific cell type. The challenge lies in developing post-processing algorithms that yield maximum overlap between lifetime images generated from the endogenous signal and ground-truth images from the exogenous antibody fluorescence signal. To properly evaluate microglia identification with endogenous NADH signal [43], colocalization analysis can be of great benefit to quantitatively analyze the overlapping areas. Besides, there is concern that when GFP is used as a marker for cellular visualization and the NADH channel is used for lifetime analysis, there is a GFP signal bleed through to the NADH channel, which can affect lifetime analysis [44]. Colocalization analysis can help us identify pixels with higher GFP bleed through [45] and recalibrate the analysis. Fortunately, ImageJ includes the colocalization analysis plugin coloc-2, which is a great candidate for our use case that can be used in conjunction with lifetime analysis.

The data was obtained from a custom multiphoton microscope built around an inverted Nikon Eclipse TE2000U and 20X (Nikon Plan Apo VC 0.75NA) objective lens. For NAD(P)H imaging, an excitation wavelength of 740 nm, emission filter centered around 457 nm (FF01-457/50, Semrock, Rochester, NY) was used. The excitation was set at 890 nm for GFP imaging, and an emission filter at 520 nm (FF01-520/35-25, Semrock, Rochester, NY) was used. FLIM data were generated using TCSPC electronics (SPC 150, Becker & Hickl, Berlin, Germany). FLIM images of 256 × 256 pixels were collected with 256-time bins and 120 s collection times.

Here, we explore the potential of FLIMLib fitting routines using FLIMJ connected with the ImageJ colocalization plugin [45]. We use a ground truth generated from CX3CR1 GFP images and use the NADH FLIM signal from the same field-of-views. The NADH FLIM data is then fitted using two-component fit using FLIMJ, and the mean lifetime image is compared with the ground truth image using the colocalization plugin [46]. The processing steps are described in Fig 3A. Fig 3C shows the overlaid image of the mean lifetime (red) and antibody intensity (green). Fig 3D shows the intensity histogram from the colocalization analysis using the coloc2 plugin, and the colocalization was ~33%. This colocalization can help us evaluate a score of overlap between antibody channel and NADH-metabolism and extend the results into analysis where a non-fluorescent antibody such as Ionized calcium-binding adaptor molecule 1 (iba1) is used [39, 46]. The detailed biology and metabolic interpretations are reported elsewhere [46].

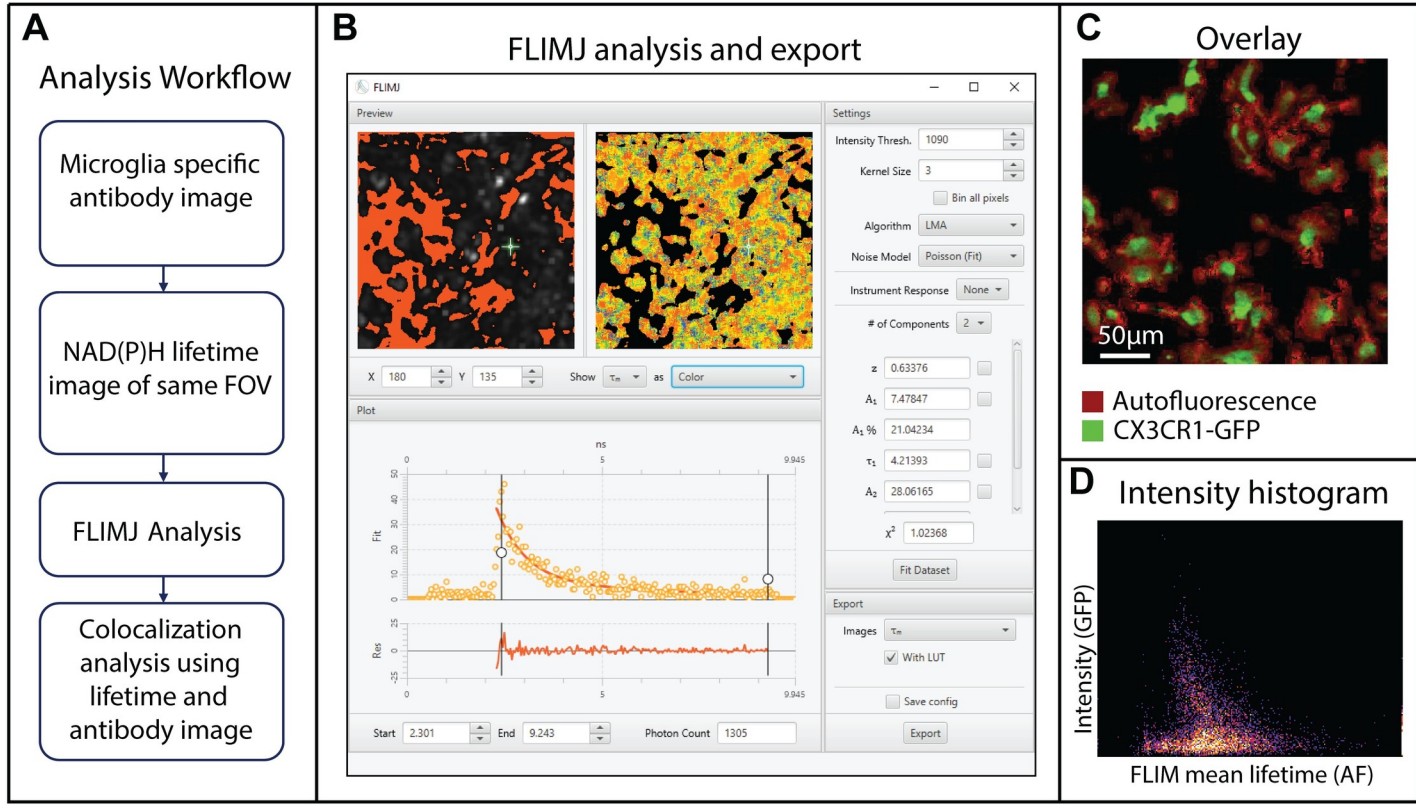

**Fig 3. Microglia colocalization analysis using NADH FLIM and CX3CR1-GFP labels** [39, 46]. **A**) The analysis workflow describing how microglia are visualized using a specific antibody, followed by NADH FLIM acquisition and FLIMJ analysis **B**) NADH FLIM data analysis using 2-component fit in FLIMJ-UI. Users can choose the intensity threshold, kernel size, fitting model, noise model, model restraints, and the number of components. The single curve fit is fast, and the "Fit Dataset" button performs fits for all the pixels. The fit result and fitting-parameters can be exported from the export tab on the lower right part of the UI. **C**) Overlaid images of antibody (green) and lifetime image (red) to show the pixels with overlapping NADH and GFP signal. **D**) Coloc2 analysis of mean lifetime and microglia antibody image.

## Method validation

We validated the lifetime value estimations of FLIMLib (see Fig 4) using a FLIM image of a fluorescence standard: fluorescein in water with a known lifetime of 4.0 ns [47]. The results were compared with the conventional fitting routine provided by the hardware vendor (SPCImage) and using the FLIMJ/FLIMLib routine. The data were analyzed using a 3x3 kernel and fit to a single component model in the LMA setting for both analyses. The vendor's routine results in an identical distribution to FLIMLib fitting results. The fitting results shown here are derived by setting the offset value to fixed numbers and fitting for the two parameters: lifetime and amplitude. The average fitting time for SPCImage and FLIMJ is identical for similar fitting parameters. There is an apparent change in speed when spatial binning is changed. FLIMJ performs spatial binning by convolving the input data with the kernel using FFT, and this operation is separate from the fitting, while SPCImage calculates the kernel every time within the fitting routine. The calculation times were computed for **SPCImage 7.4** and **flimj-ops 2.0.0** on the same computer. Neither of these comparisons used GPU optimization for testing, which could be advantageous for fitting large image datasets with homogenous fitting parameters.

We tested different fitting routines available within the flimj-ops framework (Refer to Fig 5). Two data sets were simulated for testing: 1) one component model and 2) a two-component model. The one component model was used to compare results from LMA, Phasor, and RLD methods. Fit results were plotted against the ground truth values, and a linear ~1:1 relation was seen between these three commonly used fitting methods. We also tested the phasor plots and found the phasors for single component lifetime curves fit on the universal circle. The universal phasor circle is plotted based on the fitting range. In the simulations, we used a 117 MHz universal circle to match the fitting range, instead of the more common 80 MHz plot (matched to the repetition rate of the laser used) [48]. When compared to other available commercial software bundles, two significant advantages of our library are its ability to use the Bayesian lifetime estimation method and Global fitting routines [32, 49]. We validated the **fit-Bayes** function on the same one component dataset. However, Bayesian estimates work best at low photon images and low-light FLIM experiments. For a practical comparison, we simulated

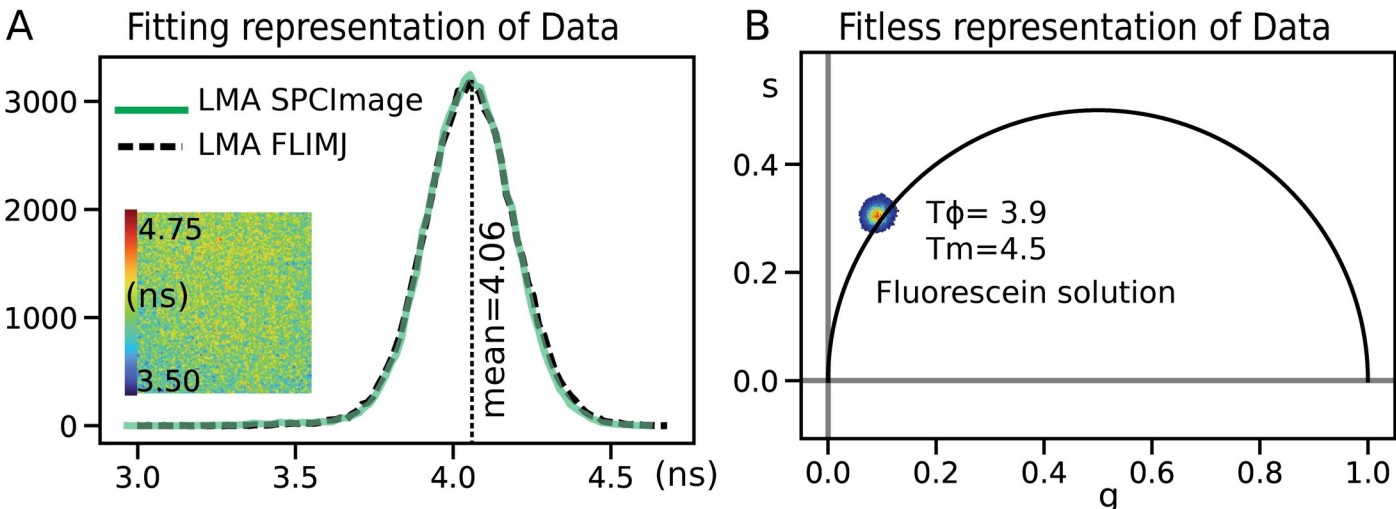

**Fig 4.** A) Validation of FLIMLib LMA lifetime estimation against hardware vendor-provided software (SPCImage) for fluorescein in water. B) The phasor plot for the data is also shown as a proof of principle of the FLIMLib Phasor function for fit-less estimation of lifetime parameters. The two parallels of lifetime histograms and phasor plots are the current laboratory standards for FLIM analysis. The phasor is plotted on a universal circle derived from the endpoints of the fit-range (117 MHz).

a noisy dataset by sampling only a fraction of the photons from the original decay curve (ground truth). We simulated nine datasets with varying the fraction of photons: 10% to 90% of total photons from the curve with an increment of 10%. We compared the results of LMA and Bayes obtained by fitting this noisy or partial data against the results from clean decay curve fit.

Both clean and noisy data were simulated on the same image, as shown in Fig 5. A sample intensity/photon distribution of 10%, 50%, and 90% data is shown in Fig 5F, along with representative decay curves from the noisy data at 4 ns. When compared with LMA, Bayes fits have a smaller difference to the clean curve fits. This can be seen in the comparison panels Fig 5G and 5H. Fig 5H shows all the sampled lifetime values (0.2 to 6.0 ns, 32 points) for all nine limited-photon data. The mean difference (mean across all pixels of the same ground-truth lifetime value) of the noisy-data lifetime estimate to the clean-data lifetime estimate is plotted as one data point. This difference is better revealed in Fig 5G, where 10% and 90% photon data are compared. Bayes fits show a low variation in comparison to LMA, but more importantly, the Bayes fit with 10% photons data is able to accurately match the characteristics of the clean curve better than LMA fits with as high as 90% photons.

For validating Global fitting, we simulated a spatially varying two-component model Refer to Fig 6). We demonstrate the fitting results from data simulated by two fluorescent species (Fig 6A and 6B) with fixed lifetime values (0.4 ns and 2.1 ns), with spatially varying fractions. These values were chosen as an example of widely used autofluorescence FLIM analyses of NAD(P)H and FAD [50, 51]. In the FLIM data shown, the top left is 100% species A and the bottom right is purely species B. The image is made of 128x128 pixels. We tested all the available fitting models on this dataset to estimate the time taken by each method (without any spatial binning). The measured timings were: **fitLMA** required ~1 second, **fitGlobal** ~0.5 seconds, **fitRLD** ~0.16 seconds, **fitPhasor** ~0.27 seconds, and **fitBayes** ~0.95 sec *(using 1-component. fitBayes is currently implemented only for 1-component analysis).*

## Discussion

In this paper, we presented FLIMJ, an open-source toolkit for curve fitting and analysis of lifetime responses. We demonstrated how it could be integrated with a variety of ImageJ workflows, including segmentation, colocalization, and cross-language analysis for Python. The use of the toolkit is also possible from other languages such as JavaScript, Groovy, or the R statistics package.

FLIMJ is powered by the FLIMLib library, which includes a range of fitting routines for lifetime data based on Levenberg-Marquardt [30], Bayesian [27, 49] as well as analysis tools such as the phasor method [25] and rapid lifetime determination method [29]. FLIMJ supports nonlinear least-squares fitting (NLS) [52] and Maximum likelihood estimation (MLE) minimization methods [53]. With support from the ImageJ Ops algorithmic framework, FLIMJ Ops can extend the usage of FLIMLib functions beyond plain curve fitting and seamlessly incorporates them with Ops-based image analysis workflows. The described analysis tools of FLIMJ can be used for a wide range of FLIM experiments. FLIM applications include simple detection of a change in lifetime due to a change in the chemical environment [54], such as viscosity [33]. More advanced experiments, such as the detection of FRET, are possible with FLIMJ. Biologically useful analysis extensions are also possible based on this core algorithmic functionality, such as global analysis, to increase the signal to noise ratio in lifetime invariant systems. Support analysis conclusions to determine the "confidence" in the fitted parameters using chi-squared maps, and model selection options to help selection of the most appropriate model for the data are all critical tools in a FLIM analysis toolbox, that are also available in FLIMJ.

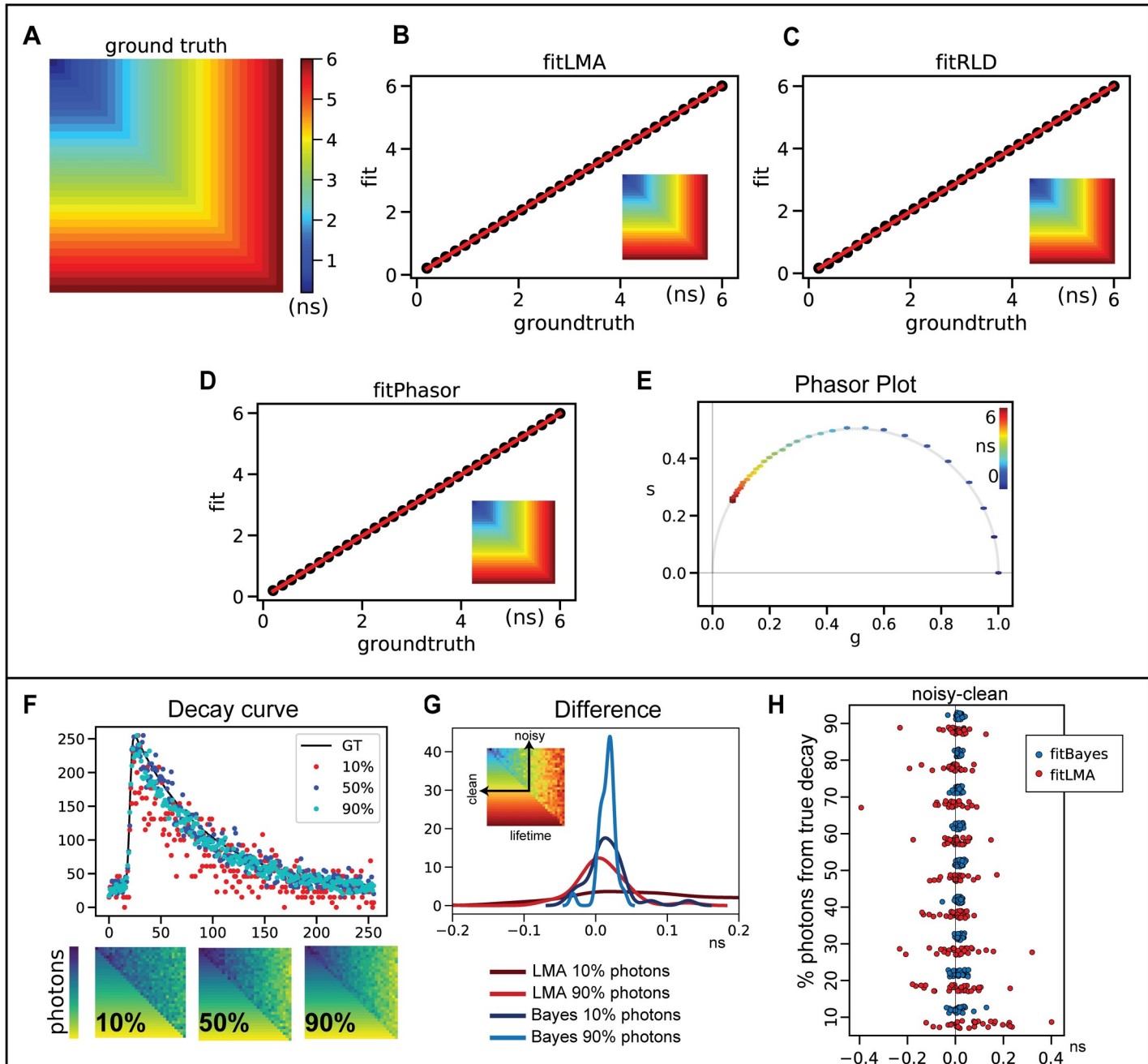

**Fig 5. FLIM Validation using a simulated dataset for 1-component fitting.** The lifetime values range from 0.2 ns to 6.0 ns that fit within the fitting range of 10 ns. A) Ground truth lifetime distribution. B) Ground truth is compared against lifetime estimates obtained using B) LMA, C) RLD, and D) Phasor. The insets in panels B-D shows the maps of lifetime estimates generated by each method. E) The phasor plot for the estimates shown in panel D. Panels F-H) shows how Bayesian fits generate a better accuracy in low-photon decay curves. F) The ground truth data was trimmed down to a fraction of total photons (10% - 90%). The data is divided diagonally half as noisy and clean. Three representative images of 10%, 50%, and 90% of total photons are compared for total photons and a sample decay curve at 4ns. A different color-map is used here to highlight that this is the photon counts and not the lifetime map. G, H) The differences between fit results of the noisy (10% - 90% photons) and clean decay curves of a range of lifetime values (0.2–6.0 ns) are presented in panel H. The LMA distribution shows a larger variance at lower photons in comparison to Bayes. Bayes fit gives a better representation to the clean curve than LMA with as low as 10% total photons. Panel G compares two representative distribution of photons: 10% and 90% for both Bayes and LMA. Bayes converges approximately four times better than LMA. The inset in panel G shows how the values are extracted for 32-lifetime value.

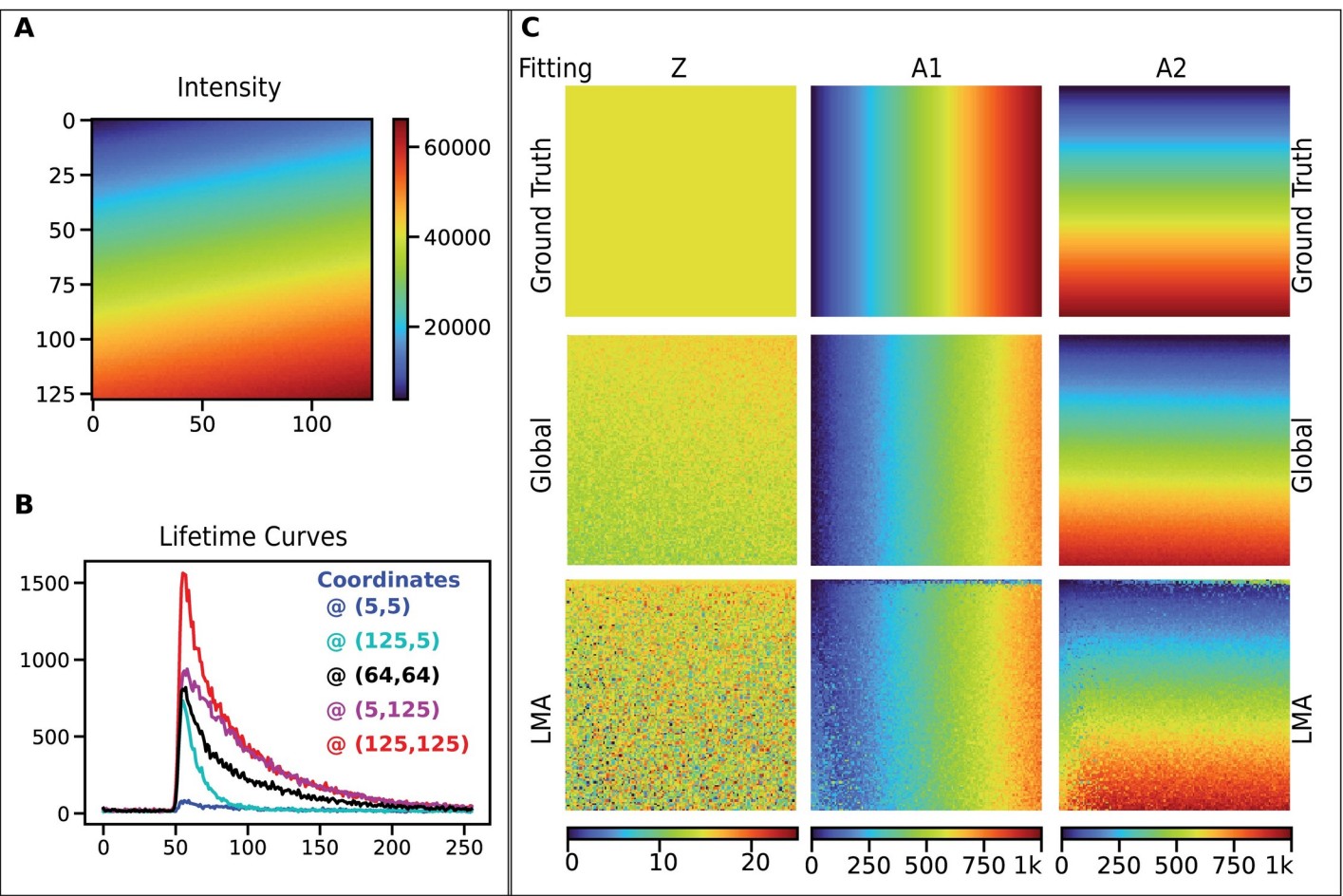

**Fig 6. FLIM Validation using simulated 2-component lifetime data.** The figure shows the simulated data with fixed lifetime values, 2.1 ns, and 0.4 ns. A) The intensity image for the simulated dataset is shown here. Note that this scale is for photons. B) Five sample lifetime curves are shown to demonstrate the variation in their intensity levels and decay rates. C) This panel compares the Global fitting routine and LMA for the two-component model. The color maps are the same between the panels of each parameter. The three parameters shown are Z (offset), A1 (amplitude of species 1), and A2 (amplitude of species 2). Both LMA and Global fitting reproduce the apparent fraction of two species, but we find that global fitting yields less noise and works twice as fast. (This dataset is provided in the SCIFIO sample datasets or https://samples.scif.io/Gray-FLIM-datasets.zip).

Although FLIMJ has successfully integrated FLIM analysis into ImageJ as an image analysis workflow, several improvements can be made. This may be mitigated by introducing global optimization algorithms before and/or after the fit. Further, although the fitting routines have been verified with data from our laboratory, more validation can be done as users start to use FLIMJ on theirs collected from different systems. We also plan to improve FLIMJ Ops and FLIMJ-UI in terms of usability. Until the current stage of development, more of the effort has been put on implementing new features than on demonstrating the existing ones. While the FLIMJ Ops landscape is being finalized, we will shift our focus towards updating and creating demonstrative tutorials using the ImageJ Wiki page or Jupyter Notebooks. Other usability improvements may include allowing saving of fitting configurations as a workspace file as in TRI2, implementing batch-fitting in the GUI, and packaging FLIMJ as standalone runnable for those without access to ImageJ. Much of the future work will also focus on further extending the functionality of FLIMJ.

Development and distribution of FLIMLib, in particular, will be aimed at the simultaneous analysis of spectral, lifetime, and polarization information that is now routinely captured in

some laboratories [20, 55, 56]. This is an area in which novel advanced analysis is needed, especially as photon numbers from biological samples are limited [57], and the addition of more dimensions of measurement (i.e. time, spectrum, polarization) can result in a very small number of photon counts per measurement channel. Although the current implementation of FLIMLib has been able to deliver a decent throughput, it can still be further optimized for speed and simplicity by depending on open-source libraries such as GSL and Boost, which yields benefit to throughput-demanding applications, including machine learning-assisted FLIM analysis. Lastly, as Fiji continues to be optimized for speed and performance and explores parallelization and possible GPU based applications, these are areas where improved FLIM analysis performance can be evaluated as well.

## Acknowledgments

We acknowledge B. Vojnovic, S. Ameer-Beg and J. Gilbey for their significant contributions to the development of the original algorithms in TRI2, J. Nedbal for the MATLAB wrapper to FLIMlib and M. Rowley and T. Coolen for the development of the Bayesian fitting algorithms. We also thank T. Gregg, M. Merrins and B DeZonia for their useful input on the original version of the program.

## Author Contributions

**Conceptualization:** Paul R. Barber, Curtis T. Rueden, Kevin W. Eliceiri.

**Formal analysis:** Dasong Gao, Paul R. Barber, Jenu V. Chacko, Md. Abdul Kader Sagar.

**Funding acquisition:** Kevin W. Eliceiri.

**Investigation:** Dasong Gao, Paul R. Barber, Jenu V. Chacko, Md. Abdul Kader Sagar, Curtis T. Rueden, Aivar R. Grislis, Mark C. Hiner, Kevin W. Eliceiri.

**Methodology:** Dasong Gao, Paul R. Barber, Md. Abdul Kader Sagar, Curtis T. Rueden, Aivar R. Grislis, Mark C. Hiner, Kevin W. Eliceiri.

**Project administration:** Kevin W. Eliceiri.

**Resources:** Paul R. Barber, Kevin W. Eliceiri.

**Software:** Dasong Gao, Paul R. Barber, Jenu V. Chacko, Md. Abdul Kader Sagar, Curtis T. Rueden, Aivar R. Grislis, Mark C. Hiner.

**Supervision:** Kevin W. Eliceiri.

**Validation:** Paul R. Barber, Jenu V. Chacko.

**Visualization:** Jenu V. Chacko.

**Writing – original draft:** Dasong Gao, Paul R. Barber, Curtis T. Rueden, Kevin W. Eliceiri.

**Writing – review & editing:** Dasong Gao, Paul R. Barber, Jenu V. Chacko, Md. Abdul Kader Sagar, Curtis T. Rueden, Mark C. Hiner, Kevin W. Eliceiri.

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
