## [Decision Letter · Decision Letter 0]

12 Oct 2020

PONE-D-20-24927

FLIMJ: an open-source ImageJ toolkit for fluorescence lifetime image data analysis

PLOS ONE

Dear Dr. Eliceiri,

Thank you for submitting your manuscript to PLOS ONE. After careful consideration, we feel that it has merit but does not fully meet PLOS ONE’s publication criteria as it currently stands. Therefore, we invite you to submit a revised version of the manuscript that addresses the points raised during the review process.

The manuscript reports a very practical open source software tool that will be of interest to readers. Please make edits or updates to the manuscript as appropriate based on the reviewers' comments below and attached to improve the clarity of the paper.

We look forward to receiving your revised manuscript.

Kind regards,

Kristen C. Maitland, Ph.D.

Academic Editor

PLOS ONE

Journal Requirements:

2.  We note that Figures [2 and 3] in your submission contain copyrighted images. All PLOS content is published under the Creative Commons Attribution License (CC BY 4.0), which means that the manuscript, images, and Supporting Information files will be freely available online, and any third party is permitted to access, download, copy, distribute, and use these materials in any way, even commercially, with proper attribution. For more information, see our copyright guidelines: http://journals.plos.org/plosone/s/licenses-and-copyright.

1.         You may seek permission from the original copyright holder of Figures [2 and 3] to publish the content specifically under the CC BY 4.0 license.

Reviewers' comments:

Reviewer's Responses to Questions

**Comments to the Author**

1. Is the manuscript technically sound, and do the data support the conclusions?

Reviewer #1: Partly

Reviewer #2: Yes

2. Has the statistical analysis been performed appropriately and rigorously? 

Reviewer #1: Yes

Reviewer #2: N/A

3. Have the authors made all data underlying the findings in their manuscript fully available?

Reviewer #1: Yes

Reviewer #2: Yes

4. Is the manuscript presented in an intelligible fashion and written in standard English?

Reviewer #1: No

Reviewer #2: Yes

5. Review Comments to the Author

Reviewer #1: The authors present a report of their validation and use cases for an important open source software tool, FLIMJ. The work is important because there are not many easy-to-use lifetime image analysis tools integrated with broader image analysis tools such as those that can be found in the FIJI ecosystem.

While it does not appear the authors have developed novel analysis algorithms (the toolkit is built using the previously existing FLIMLib library and Scijava components) or answered an original research question using FLIMJ (demonstration of the software is given by way of previous studies), the reviewer believes that the transformation of the FLIMLib backend into a user-friendly tool is of significant relevance to the PLOS One audience, not to mention an endeavor requiring significant investment of time and expertise. The combined result is something new.

The article requires significant work to make it more readable and compelling.

I recommend the article be accepted for publication after the authors address the items in the attachment.

Reviewer #2: The authors presented an open source software toolbox for Fluorescence Lifetime Imaging (FLIM) applications. This toolbox, called FLIMJ, is an extension to the widely used FIJI-ImageJ software, which will facilitate curve fitting and analysis of fluorescence lifetime measurements. Additionally to ImageJ, the flexibility of this software allows effortless integration with a variety programming platforms such as Python, JavaScript, Groovy, and R. Some key features include different decay fitting routines, segmentation, colocalization, and cross-language analysis, which are complemented with the set of image processing tools from ImageJ. The scientific community using FLIM may potentially be benefited by using this free access toolkit. Below there are some comments that might help to improve the publication of this manuscript.

1. The proposed FLIM tool kit will greatly benefit those whose research is based on analysis of FLIM features and not on development of methods for FLIM analysis. However, there is a community of researches using machine learning libraries on FLIM data that are not available for ImageJ such as the MATLAB Machine learning toolbox, TensorFlow, and PyTorch - to mention some. Currently, Python is gaining popularity through the microscopy community due to the vast availability scientific toolboxes. The authors stated that FLIMJ can be integrated to Python.

However, a Jupyter notebook that accesses ImageJ ops through the PyImageJ interface to invoke FLIMJ ops is required. It would be also very beneficial to have a direct FLIMLib library for Python that can be directly installed via Conda or PIP to straightforwardly connect it with the statistical and machine learning libraries for Python without interfacing PyImageJ.

2. ImageJ can read different image file formats such as TIFF, NIfTI, DICOM, HDF5, etc. However, to my knowledge, it cannot read proprietary file formats of commercial FLIM software such as Becker & Hickl (B&H) and PicoQuant. Does FLIMJ provide integration of file formats from commercial FLIM systems?

3. Could the authors please talk about the reproducibility of image processing from software of commercial FLIM systems, since commercial software uses proprietary algorithms for decay fitting and image segmentation that might differ from the ones provide by FLIMJ?

4. Some research studies require analysis of 3D volumetric FLIM images in time (4D imaging), and in some cases longitudinal multispectral volumetric FLIM imaging (5D imaging). Could the authors please discuss how FLIMJ would provide support these kind of data without using programming since the idea of using ImageJ is to avoid creating your own image processing scripts?

5. A significant number of researchers working on FLIM applications use open source software options such as FLIMfit. How easy/hard would be the transition from FLIMfit to FLIMJ?

6. Is FLIMJ including basic FLIM calculations such as optical redox ratio without doing it manually in ImageJ?

6. PLOS authors have the option to publish the peer review history of their article (what does this mean?). If published, this will include your full peer review and any attached files.

Reviewer #1: No

Reviewer #2: No

---

## [Author Response · Author response to Decision Letter 0]

27 Nov 2020

Please see the attached document for detailed response to the reviewers.

---

## [Decision Letter · Decision Letter 1]

15 Dec 2020

FLIMJ: an open-source ImageJ toolkit for fluorescence lifetime image data analysis

PONE-D-20-24927R1

Dear Dr. Eliceiri,

We’re pleased to inform you that your manuscript has been judged scientifically suitable for publication and will be formally accepted for publication once it meets all outstanding technical requirements.

Kind regards,

Kristen C. Maitland, Ph.D.

Academic Editor

PLOS ONE

Additional Editor Comments (optional):

Reviewers' comments:

Reviewer's Responses to Questions

**Comments to the Author**

1. If the authors have adequately addressed your comments raised in a previous round of review and you feel that this manuscript is now acceptable for publication, you may indicate that here to bypass the “Comments to the Author” section, enter your conflict of interest statement in the “Confidential to Editor” section, and submit your "Accept" recommendation.

Reviewer #1: All comments have been addressed

Reviewer #2: All comments have been addressed

2. Is the manuscript technically sound, and do the data support the conclusions?

Reviewer #1: Yes

Reviewer #2: Yes

3. Has the statistical analysis been performed appropriately and rigorously? 

Reviewer #1: Yes

Reviewer #2: Yes

4. Have the authors made all data underlying the findings in their manuscript fully available?

Reviewer #1: Yes

Reviewer #2: Yes

5. Is the manuscript presented in an intelligible fashion and written in standard English?

Reviewer #1: Yes

Reviewer #2: Yes

6. Review Comments to the Author

Reviewer #1: The authors' edits to the manuscript and figures have improved the reader's ability to quickly and clearly understanding the benefits of FLIMJ. Thank you for creating this tool for the imaging and biosciences communities.

Reviewer #2: The authors have addressed all my concerns and therefore I support publication without further

changes.

7. PLOS authors have the option to publish the peer review history of their article (what does this mean?). If published, this will include your full peer review and any attached files.

Reviewer #1: No

Reviewer #2: No

---

## [Editor Report · Acceptance letter]

18 Dec 2020

PONE-D-20-24927R1 

FLIMJ: an open-source ImageJ toolkit for fluorescence lifetime image data analysis 

Dear Dr. Eliceiri:

I'm pleased to inform you that your manuscript has been deemed suitable for publication in PLOS ONE. Congratulations! Your manuscript is now with our production department. 

Kind regards, 

on behalf of

Dr. Kristen C. Maitland 

Academic Editor

PLOS ONE